# Influence of Simulated In Vitro Gastrointestinal Digestion on the Phenolic Profile, Antioxidant, and Biological Activity of *Thymbra spicata* L. Extracts

**DOI:** 10.3390/antiox11091778

**Published:** 2022-09-09

**Authors:** Farah Diab, Mohamad Khalil, Giulio Lupidi, Hawraa Zbeeb, Annalisa Salis, Gianluca Damonte, Massimo Bramucci, Piero Portincasa, Laura Vergani

**Affiliations:** 1Department of Earth, Environment and Life Sciences (DISTAV), University of Genova, Corso Europa 26, 16132 Genova, Italy; 2Clinica Medica “A. Murri”, Department of Biomedical Sciences and Human Oncology, Medical School, University of Bari “Aldo Moro”, 70124 Bari, Italy; 3School of Pharmacy, University of Camerino, 62032 Camerino, Italy; 4Department of Experimental Medicine (DIMES), University of Genova, Viale Benedetto XV 1, 16132 Genova, Italy; 5Centre of Excellence for Biomedical Research (CEBR), University of Genova, Viale Benedetto XV 9, 16132 Genova, Italy

**Keywords:** *Thymbra spicata* L. extracts, in vitro gastrointestinal digestion, phytochemical characterization, antioxidant capacity, cytotoxic effect

## Abstract

Plants or plant extracts are widely investigated for preventing/counteracting several chronic disorders. The oral route is the most common route for nutraceutical and drug administration. Currently, it is still unclear as to whether and how the pattern of phenolic compounds (PCs) found in the plants as well as their bioactivity could be modified during the gastrointestinal transit. Recent studies have revealed antioxidant and anti-steatotic properties of Thymbra spicata. Here, we investigated the possible loss of phytochemicals that occurs throughout the sequential steps of a simulated in vitro gastrointestinal (GI) digestion of aqueous and ethanolic extracts of aerial parts of *T. spicata*. Crude, digested, and dialyzed extracts were characterized in terms of their phenolic profile and biological activities. Total contents of carbohydrates, proteins, PCs, flavonoids, and hydroxycinnamic acids were quantified. The changes in the PC profile and in bioactive compounds upon the simulated GI digestion were monitored by HPLC–MS/MS analysis. The antioxidant activity was measured by different spectrophotometric assays, and the antiproliferative potential was assessed by using three representative human cancer cell lines. We observed that the simulated GI digestion reduced the phytochemical contents in both aqueous and ethanolic *T. spicata* extracts and modified the PC profile. However, *T. spicata* extracts improved their antioxidant potential after digestion, while a partial reduction in the antiproliferative activity was observed for the ethanolic extract. Therefore, our results could provide a scientific basis for the employment of *T. spicata* extract as valuable nutraceutical.

## 1. Introduction

Dietary phytochemicals are found abundantly in fruits, vegetables, grains, plant-based foods, and beverages [1]. Consumption of phytochemicals plays a main role in healthcare by preventing many chronic diseases including non-alcoholic fatty liver disease (NAFLD) [2], cardiovascular disease [3], neurodegenerative diseases [4], and some types of cancer [5]. For this reason, extracts from plants or plant parts have been largely tested to develop new functional foods for preventing/counteracting many chronic disorders [6]. Phenolic compounds (PCs) are the most abundant phytochemicals in many edible and medicinal plants, and they are the main responsible agents for the beneficial effects, especially the defense against oxidative stress [7]. PCs include numerous varieties of compounds classified into flavonoids and non-flavonoids: flavonoids include flavonols, flavones, flavan-3-ols, flavanones, and anthocyanins; non-flavonoid compounds include phenolic acids, volatile phenols, stilbenes, lignans, and coumarins [8].

Importantly, the bioavailability of several plant extracts as a source of PCs is likely affected by changes occurring during the gastrointestinal (GI) transit. Foods and nutraceuticals introduced by the oral route undergo digestive processes throughout GI compartments and cross physiological barriers that are able to influence their delivery [9]. Indeed, the main challenges for bioactive compounds are the rate and degree of absorption, as well as their solubility, stability, and permeability across the mucosal and intestinal barriers [10]. Moreover, metabolites can show completely different bioavailability compared to the parental phenolic compounds due to the physiological environment and cofactors [11,12]. Experimental approaches using in vitro GI models can overcome difficulties associated with human studies that are often poorly reproducible and comparable, expensive, time-consuming, and might generate ethical issues, depending on the study design and food being tested [13]. In fact, several digestion methods have been proposed in the literature review, often differing in the applied conditions. To give an example, the origin of the used enzymes (porcine, rabbit, or human); the environmental factors (pH, ionic strength, and digestion time); and other parameters such as the presence of phospholipids, digestive emulsifiers vs. their mixtures (e.g., pancreatin and bile salts), and the ratio of food bolus to digestive fluids, which alter enzyme activity, may considerably alter the results. While modifying some of these parameters with a possible and major impact on the matrix release or digestibility of some compounds, we were concerned with applying a standardized and practical simulated in vitro GI digestion method based on physiologically relevant conditions that can be applied for various endpoints and may be amended to accommodate further specific requirements mainly developing a more accurate in vitro human digestion model, taking into consideration the intestinal microbiota presence and conditions.

*Lamiaceae* is a family of mostly shrubs and herbs with a wide distribution worldwide, especially in the Mediterranean basin [14]. In this family, *Thymbra spicata* L., locally known as “Za’atar”, is employed in the folk cookery (as salad or tea infusion), but also in traditional medicine, mainly for its antimicrobial and antiseptic properties [6]. Recent studies have revealed several beneficial properties of *T. spicata* L. leaves such as antioxidant, hypocholesterolemic, and anti-steatotic activities [15], as well as anti-inflammation [16], anti-proliferative, and pro-apoptotic [17] potential. The abundancy of PCs in *T. spicata* L. leaves including phenolic acids (rosmarinic acid), phenolic monoterpenoids (carvacrol, thymol), and flavonoids (both glycosides and aglycones) stand behind the wide array of its pharmacological activities [18,19]. The beneficial effects of *T. spicata* L. as herbal medicine or nutraceutical preparation might be modified during the GI transit where the bioconversion is elicited by low gastric pH, digestive enzymes, and the microbiota [11].

In this context, our study aimed to assess if and how two different extracts from *T. spicata* L. aerial parts were modified after applying a simulated in vitro GI digestion method. The extracts before and after digestion were characterized for their composition in bioactive compounds and their antioxidant potential. The biological effects were assessed by cellular studies focusing on the antiproliferative capacity on different cancer cell lines.

## 2. Materials and Methods

### 2.1. Reagents and Enzymes

All reagents otherwise indicated, including enzymes, were purchased from Sigma- Aldrich Corp. (Milan, Italy). All reagents were of analytical purity.

#### Sources and Activities of Enzymes

α-Amylase from human saliva (A0521-500 units/mg). α-Amylase catalyzes the hydrolysis of α-1,4 glycosidic linkage in oligosaccharides.Pancreatin from porcine pancreas (P3292-100G). Pancreatin contains enzymatic components including trypsin, amylase and lipase, ribonuclease, and protease, produced by the exocrine cells of the porcine pancreas.Pepsin from pig gastric mucosa (≈2500 units/mg protein). Pepsin is an aspartic endproteinase used for the unspecific hydrolysis of proteins and peptides in acidic media.

### 2.2. Plant Collection

Aerial parts of *Thymbra spicata* L. were collected from flowering plants growing in “Maarakeh”, South Lebanon. Voucher specimen (L1.125/1) was authenticated by Prof. G. Tohme (CNRS, Beirut, Lebanon) and was kept in the Herbarium of the Botanical Department-Lebanese University (Beirut, Lebanon). The in vitro gastrointestinal (GI) digestion of *T. spicata* aerial parts was performed according to protocol [20] with slight modifications in order to sequentially simulate the mouth, stomach, and small intestine digestion. The composition of buffers is reported in Table 1.

### 2.3. In Vitro Simulated Digestion

#### 2.3.1. Oral Digestion

To mimic the oral digestion, 25 g of *T. spicata* dried and powdered aerial parts were mixed with 25 mL SSF and 3 mL (stock 75 U/mL) α-salivary amylase (from human saliva), 0.2 mL CaCl_2_, and 5.8 mL distilled H_2_O and then incubated for 2 min at 37 °C on a magnetic stirrer.

#### 2.3.2. Gastric Digestion

To mimic the gastric digestion, 40 mL SGF, 7 mL pepsin (stock 25,000 U/mL), 0.03 mL CaCl_2_, and 3 mL distilled H_2_O were added to the oral outcome and the pH was lowered to 3.0 by HCl; the mixture was incubated for 2 h at 37 °C on a magnetic stirrer, and the pH was checked regularly.

#### 2.3.3. Intestinal Digestion

To mimic the intestinal digestion, 50 mL of gastric outcome was mixed with 50 mL of SIF, 20 mL of pancreatin (stock 100 U/mL), 10 mL bile salt (stock 10 mM), 0.024 mL CaCl_2_, 6 mL distilled H_2_O, and 0.7 mL of 1 M HCl to neutralize the pH to 7.0.

#### 2.3.4. Extract Preparation

The obtained mixture was incubated for 2 h at 37 °C on a magnetic stirrer. Then, the mixture was heated to 90 °C for 10 min to inactivate the enzymes used in the digestion process. At the end, the samples were centrifuged for 20 min at 7000 rpm. The pellet was incubated with ethanol (96%) at room temperature for 24 h with agitation. The solution was centrifuged for 20 min at 7000 rpm, obtaining a precipitate (TE) that was discharged, and the ethanol in the digested ethanolic extract (TE-dig) was removed using a rotavapor before the lyophilization of the residue. The supernatant obtained from the digestion process was divided into two parts. One part was lyophilized, obtaining the crude digested aqueous extract (TW-dig); the other one was dialyzed with membrane cut-off 3.5 kDa (Spectra/Por molecularporous membrane tubing, Thermo Fisher Scientific, Milan, Italy) against 250 mL of water for 24 h at 4 °C to separate the low molecular weight (mw) fraction (<3.5 kDa) and the high mw fraction (>3.5 kDa). The solutions inside the dialysis tube (>3.5 kDa) and out of the tube (<3.5 kDa) were lyophilized.

To prepare the undigested aqueous and ethanolic extracts (TW and TE, respectively), the same procedure was followed for extraction without the first part of enzymatic digestion. The detailed scheme of in vitro digestion and the obtained extracts and fractions is illustrated in Figure 1.

### 2.4. Total Carbohydrate Content (TCC)

TCC was determined by the phenol-sulfuric acid colorimetric method [21]. Briefly, 0.5 mL of sample (1 mg/mL) was mixed with 0.5 mL 5% aqueous phenol and 2 mL of H_2_SO_4_ (96%). After incubation for 30 min at room temperature, the absorbance was read at 320 nm using a UV–VIS microplate reader (FLUOstar Optima, BMG Labtech, Ortenberg, Germany). The results were derived from a glucose calibration curve (0–200 µg/mL). Values are expressed as µg/mg extract.

### 2.5. Total Protein Content (TPrC)

The protein content was determined by Bradford colorimetric method, using bovine serum albumin (BSA) as standard [22]. Briefly, 0.5 mL from each extract (1 mg/mL) was mixed with 0.5 mL of Bradford reagent; after 30 min incubation, the absorbance was measured at 595 nm using a UV–VIS microplate reader (FLUOstar Optima, BMG Labtech, Ortenberg, Germany). Data are expressed as µg/g.

### 2.6. Total Phenol Quantification (TPC)

TPC was determined using the Folin–Ciocalteu method [23]. Briefly, 25μL aliquots of sample (1 mg/mL) were incubated with 125 μL of 10% (*w*/*v*) Folin–Ciocalteu reagent for 5 min; after adding 125 µL of Na_2_CO_3_ (10% *w*/*v*), the sample was incubated for 30 min in darkness at room temperature, and the absorbance was read at 320 nm using a UV–VIS microplate reader (FLUOstar Optima, BMG Labtech, Ortenberg, Germany). The results were derived from a gallic acid calibration curve (0–1000 ug/mL) prepared from a stock solution (1 mg/mL in ethanol). Values are expressed as mg of gallic acid equivalents (GAE) per gram of dried weight extract (mg of GAE/g extract).

### 2.7. Total Flavonoid Quantification (TFC)

TFC was determined using the aluminum chloride colorimetric method [24]. Briefly, a 1 mL aliquot of sample (1 mg/mL) was mixed with 0.2 mL of 10% (*w*/*v*) methanolic AlCl_3_ solution, 0.2 mL (1 M) potassium acetate, and 5.6 mL distilled H_2_O. After incubation at room temperature in darkness for 30 min, the absorbance was read at 320 nm using a UV–VIS microplate reader. The results were derived from a calibration curve of quercetin (0–200 μg/mL) prepared from a stock solution (5 mg/mL in methanol). Values are expressed as mg of quercetin equivalent (QE) per gram of dried weight extract (mg of QE/g extract).

### 2.8. Total Hydroxycinnamic Acid Content (THAC)

HCA was determined using the method by Custódio et al. [25]. Briefly, in a 96-well plate, 20 µL of sample (5 mg/mL) was mixed with 20 µL of 95% ethanol containing 0.1% HCl. After the addition of 160 µL of 2% HCl and 10 min incubation, the absorbance was read at 320 nm using a UV–VIS microplate reader. The results were derived from a calibration curve of rosmarinic acid (0–500 µg/mL) prepared from a stock solution (1 mg/mL in ethanol). Values are expressed as mg of rosmarinic acid equivalents (RAE) per gram of dried weight extract (mg of RAE/g extract).

### 2.9. HPLC–MS Analysis

High-performance liquid chromatography coupled with tandem mass spectrometry (HPLC–MS/MS) was performed using an Agilent 1100 HPLC-MSD Ion Trap XCT system, equipped with an electrospray ion source (HPLC-ESI-MS) (Agilent Technologies, Santa Clara, CA, USA). Separation of extracts was performed on a Jupiter C18 column 1 × 150 mm with 3.5 μm particle size (Phenomenex, Torrance, CA, USA). As eluents, we used water (eluent A) and MeOH (eluent B), both added with 0.1% formic acid. The gradient employed was 15% eluent B for 5 min, linear to 100% eluent B in 35 min, and finally hold at 100% eluent B for another 5 min. The flow rate was set to 50 μL/min with a column temperature of 30 °C. The injection volume was 8 μL. Ions were detected in the positive and negative ion mode, in the *m*/*z* 100–800 range, and ion charged control with a target ion value of 100,000 and an accumulation time of 300 ms. A capillary voltage of 3300 V, nebulizer pressure of 20 psi, drying gas of 8 L/min, dry temperature of 325 °C, and 2 rolling averages (averages: 5) were the parameters set for the MS detection. MS/MS analysis was conducted using an amplitude optimized time by time for each compound. From the chromatograms, the percentage of PC for each extract was calculated on the basis of the peak area.

### 2.10. Radical Scavenging Activity Assays

The radical scavenging activity was measured using the 1,1-diphenyl-2-picrylhydrazyl (DPPH) method [26]. In a 96-multiwell plate, 50 µL aliquot of sample (0–2 mg/mL) or of the standard Trolox (0–100 mg/mL) was added to 200 µL of DPPH solution (0.1 mM in methanol). After incubation in darkness for 30 min at 37 °C, the absorbance was measured at 490 nm using a UV–VIS microplate reader against DPPH solution as a blank. Values are expressed as half maximal inhibitory concentration IC50 (µg/mL) and Trolox equivalent (µg TE/mg dry extract).

The radical cation scavenging activity of each extract was measured using the 2-2′-azino-bis (3-ethylbenzo-thiazoline-6-sulphonate) diammonium salt (ABTS) method [27]. In a 96-multiwell plate, 50 µL aliquot of sample (0–2 mg/mL) was added to 200 µL of ABTS solution (5 mM). ATBS solution was prepared by oxidizing ABTS with MnO_2_ in distilled water for 30 min in the dark, and then the solution was filtered through filter paper. After 20 min incubation in darkness at room temperature, the absorbance was determined at 734 nm using a UV–VIS microplate reader against ABTS solution as a blank. Values are expressed as half maximal inhibitory concentration IC50 (µg/mL) and Trolox equivalent (µg TE/mg dry extract).

### 2.11. Ferric Reducing Antioxidant Power (FRAP) Assay

The reducing power was evaluated according to the ferric reducing antioxidant power (FRAP) assay [28]. In a 96-multiwell plate, 25 µL aliquot of sample (0–2 mg/mL) or of standard Trolox (0–100 µg/mL) was added to 175 µL of FRAP working solution containing 300 mmol/L acetate buffer (pH 3.6), 20 mmol/L ferric chloride, and 10 mmol/L TPTZ (2,4,6-tri (2-pyridyl)—S-triazine) made up in 40 mmol/L HCl. The three solutions were mixed at a 10:1:1 ratio (*v*:*v*:*v*). The mixture was incubated in darkness for 30 min at 37 °C and then the absorbance was determined at 593 using a UV–VIS microplate reader against FRAP solution as a blank. Values are expressed as Trolox equivalent (µg TE/mg dry extract).

### 2.12. Cell Culture

The human cancer cell lines MDA-MB-231 (breast adenocarcinoma), A375 (Melanoma), and HCT116 (colorectal carcinoma) were gently supplied from Prof. Bramucci (Laboratory of Physiology, University of Camerino). The cancer cells were routinely maintained in Dulbecco’s modified Eagle’s minimum essential medium (DMEM) or in RPMI-1640 (Sigma-Aldrich, Beirut, Lebanon) supplemented with 10% heat-inactivated fetal bovine serum (FBS), 2 mM glutamine, and 1% P/S at 37 °C in a humidified incubator containing 5% CO_2_.

### 2.13. Cell Proliferation Assay

The cytotoxicity of *T. spicata* extracts was assessed by the 3-(4,5-dimethylthiazol-2-yl)-2,5-diphenyltetrazolium bromide (MTT) method [29]. Stock solution (50 mg/mL) of extracts were prepared in dimethyl sulfoxide (DMSO) or in sterile distilled water. In addition, the pure Carvacrol was used as positive control. Briefly, cells were seeded in a 96-well plate (104 cells per well), and after 24 h, they were treated with increasing concentrations (0, 50, 100, and 200 µg/mL) of each extract for 24 h. At the end, 20 µL of MTT reagent (5.0 mg/mL) was added, and the mixture was incubated for 3 h at 37 °C. After removing the unreacted MTT dye, 100 µL DMSO was added to solubilize purple formazan crystals, and the absorbance was recorded at 570 nm. The IC50 value (concentration that causes 50% growth inhibition) was estimated as that leading to 50% absorbance decrease as compared to the control. Cell viability was expressed in percentage with respect to the control.

### 2.14. Quantification of ROS Production

2′,7′-Dichlorodihydrofluorescein diacetate (H_2_DCF-DA; molecular probe) was employed to assess ROS generation [30]. Briefly, cells were seeded on a 96-well plate (105 cells/mL) and incubated overnight. After the treatments, cells were washed twice with PBS and then incubated with 10 µM of H2DCF-DA (in PBS) for 30 min at 37 °C. Then, ROS production level was measured fluorometrically using a microplate reader (lex = 495 nm; lem = 525 nm).

### 2.15. Quantification of Nitrite/Nitrate Levels

The nitric oxide NOx (nitrites and nitrates) level was measured by spectrophotometric measurement using the Griess reaction [31]. Briefly, 105 cells/mL were seeded on a 96-well plate and incubated overnight. After the treatments, NOx level in the medium was calculated using NaNO_2_ as a standard curve. Spectrophotometric analyses were performed at 546 nm using a microplate reader.

### 2.16. Statistical Analysis

All results were expressed as mean ± SD of at least three independent experiments. GraphPad Prism 8.0.1 software was used for statistical evaluation. Comparisons between different conditions were performed using ANOVA with Tukey’s post-test. Difference between percentages was calculated by chi-squared test. The possible correlation between the measured parameters was tested by a two-tailed Pearson’s correlation coefficient analysis. All statistical analysis were performed by GraphPad Software Prism 8.0.1, Inc. (San Diego, CA, USA).

## 3. Results

### 3.1. Characterization of T. spicata Extracts before and after Simulated Digestion

We characterized the aqueous and ethanolic extracts from *T. spicata* aerial parts before (TW and TE) and after (TW-dig and TE-dig) the simulated digestion. For TW, we also assessed the low (<3.5 kDa) and the high (>3.5 kDa) mw fractions obtained by dialysis.

The aqueous and ethanolic crude extracts exhibited a similar content of carbohydrates. The simulated digestion significantly reduced the TCC in the ethanolic extract (from 26.4 in TE to 19.2 µg/mg in TE-dig), without affecting the aqueous extract. However, upon simulated digestion, we observed a different distribution of carbohydrates between the two fractions: TCC was reduced in the high mw fraction (from 40.4 in TW to 17.6 µg/mg in TW-dig) and increased in the low mw fraction (from 25.2 in TW to 33 µg/mg in TW-dig) (Figure 2A). Moreover, the protein content was roughly similar in the crude extracts, and the simulated digestion did not affect it considerably. However, the aqueous extract showed a redistribution of the protein content between the two mw fractions, leading to a TPrC reduction in the high mw fraction (from 10.8 in TW to 2.4 µg/g in TW-dig) and an increase in the low mw fraction (from 22.7 in TW to 33.9 µg/g in TW-dig) (Figure 2B).

As expected, the ethanolic extract was richer in phenolic compounds compared to the aqueous one (353 vs. 201.4 mg GAE/g). After simulated digestion, the TPC significantly decreased in both the ethanolic (250 mg GAE/g) and the aqueous (138.9 mg GAE/g) extracts. For the aqueous extract, the simulated digestion reduced the TPC in the high mw fraction (from 148.7 to 81.6 mg GAE/g). By contrast, the aqueous extract was richer in flavonoids than the ethanolic extract (172.88 vs. 123.84 mg QE/g).

After simulated digestion, the TFC significantly decreased in both the ethanolic (to 85.18 mg QE/g) and aqueous (to 111.26 mg QE/g) extracts, as well as in both the high (from 117.03 to 25.75 mg QE/g) and low mw fractions (from 278.13 to 236.24 mg QE/g) (Figure 2D). The hydroxycinnamic acid content was higher in the ethanolic than in the aqueous extract (89.2 vs. 35.4 mg RAE/g, respectively). The simulated digestion reduced the THAC in the crude ethanolic extract (76.5 mg RAE/g in TE-dig), while in the aqueous extract, the digestion redistributed the THAC between the two mw fractions (from 101.6 to 95.2 mg RAE/g for low mw fraction, and from 31.4 to 17.8 mg RAE/g in the high mw fraction) (Figure 2E).

### 3.2. HPLC–MS Characterization of the Phenolic Compounds

Both the extracts were characterized by HPLC–MS/MS analysis before and after digestion (Figure 3). In the ethanolic extract, we detected 14 PCs in both the undigested and digested preparations. The most abundant PCs were monoterpenoic phenols (carvacrol), polyphenolic acids (rosmarinic acid), flavonoids, and their derivatives (rutin, thymusin, and eriodictyol derivative, etc.). The aqueous extract contained less PCs; 19 PCs were detected in both the undigested and digested preparations, which can be classified into three main groups: phenolic acids, phenolic monoterpenoids, and flavonoids. Carvacrol is the most abundant PC in the ethanolic extract (34.8% in TE and 52.9% in TE-dig), rosmarinic acid in TW (57.4%), and salvalonic acid in TW-dig (42.3%).

To compare the chromatograms of the two extracts, we normalized them for their TPC. The analysis revealed some differences in the percentages of the major PCs (Table 2). The simulated digestion led to an enrichment in carvacrol abundance in the ethanolic extract (from 34.8 to 52.9%; *p* ≤ 0.01) and to a reduction in rosmarinic acid abundance in the aqueous extract (from 57.4% to 18.8%; *p* ≤ 0.01). In TW, the reduction in rosmarinic acid was almost balanced by the increase in salvalonic acid (from 19.5% to 42.3%; *p* ≤ 0.01). Moreover, a redistribution in phenols, flavonoids, and hydroxycinnamic acids was observed between the two mw fractions, with the low mw fraction being enriched in both TPC (from 1.53 to 2.06) and TFC (from 1.61 to 2.12), balanced by a reduction in the high mw fraction of TFC (from 0.68 to 0.23) and THAC (from 0.89 to 0.54) (Figure 4).

### 3.3. Effect of Simulated Digestion on Antioxidant Proprieties

The antioxidant potential of each extract before and after the simulated digestion was evaluated by three different spectrophotometric assays. A higher antioxidant activity was observed for the ethanolic extract compared to the aqueous one. Both extracts showed significant changes in the antioxidant potentials upon digestion, as well as between the mw fractions obtained by dialysis before and after digestion (Table 3A).

After normalizing the Trolox equivalent values for the TPC in each extract, we appreciated a higher antioxidant activity for the digested extracts than for the crude ones. Briefly, in the ethanolic extract, DPPH increased from 0.25 (TE) to 0.37 (TE-dig), and in the aqueous extract from 0.44 (TW) to 0.60 (TW-dig).

Regarding the FRAP assay, ethanolic extract showed an increase in the reducing capacity after digestion from 0.26 (TE) to 0.35 (TE-dig). Conversely, the ABTS assay of the aqueous extract showed a reduction in the high mw fraction from 3.06 (TW-dig > 3.5 kDa) to 1.68 (TW-dig > 3.5 kDa) (Table 3A).

Finally, the correlation analysis between the phytochemical contents (in terms of TPC, THAC, and TFC) and the three antioxidant activities (evaluated as DPPH, ABTS, and FRAP) for all the extracts showed a significant and strong correlation between DPPH and TPC (r^2^ = 0.7612). Moreover, a good correlation was calculated between FRAP and the phytochemical contents: FRAP and TFC (r^2^ = 0.8952), FRAP and THAC (r^2^ = 0.8913), and FRAP and TPC (r^2^ = 0.7755) (Table 3B). These results indicate that the phenolic compounds contained in the extracts are the major contributor for the antioxidant capacity.

### 3.4. In Vivo Effects: Cytotoxic Activity and Oxidative Stress in Cancer Cells

The anti-proliferative effects of the different extracts before and after digestion were assessed using three human cancer cell lines representative of the most common human cancers, i.e., MDA-MB 231, HCT116, and A375 cells (Figure 5). No significant cytotoxic effects were observed for the aqueous extract (data not shown). By contrast, all the ethanolic extracts, both undigested and digested, significantly reduced the cell viability of all cancer cell lines in a concentration-dependent manner. Carvacrol is the most abundant component of the ethanolic extract, and it was employed as a positive control. After 24 h of treatment at the highest concentration (200 µg/mL), both the extracts (TE and TE-dig) and carvacrol dramatically reduced the cell viability in MDA-MB 231 (to 20.1%, 28.8%, and 17.9%; respectively), in HCT116 cells (to 22.3%, 41.1%, and 7.8%, respectively), and in A375 cells (to 38.2%, 26%, and 21.6%, respectively). From the IC50 values listed in Table 4, we can deduce that carvacrol is the most cytotoxic agent for A375 cells at 24h (IC50 of about 19.912 µg/mL), as well as for MDA-MB 231 cells (IC50 about 23.278 µg/mL) and HCT 116 cells (IC50 about 59.625 µg/mL), and other intermediate inhibitory effects have been exhibited on the other cell lines.

In attempt to decipher the mechanisms sustaining the cytotoxic effects, we assessed the oxygen and nitrogen radical production. A dose-dependent increase in ROS production was detected in the breast cancer cell line MDA-MD 231 treated with either the crude or digested extracts. At the highest concentration (200 µg/mL), ROS production increased to +158.1% for TE-dig and +154.1% for TE, but carvacrol was more efficient in inducing ROS production (+238.8%). Similar results were recorded for the colon cancer cell line. 

On HCT116, carvacrol stimulated ROS production (+253.8%) more than the crude and digested extracts (TE +151.9% and TE-dig +163.6%) at the highest concentration. Moreover, for the melanoma cancer cell A375, carvacrol stimulated ROS production (+293.4%) more than the crude and digested extracts (+191.7% for TE-dig and +142.7% for TE) at the highest concentration (Figure 6A).

An opposite trend was observed in terms of NO release. In this case, the extracts, at the highest concentration (200 µg/mL), were stronger than carvacrol in triggering the NO release in MDA-MB231 (+283.% for TE-dig, +310.4% for TE, and +251.6% for CVL), as well as in HCT116 and A375 cells, where the release was maximum for TE (+242.7% and +292.6%, respectively) followed by TE-dig (+190.4% and +250.4%, respectively) and by CVL (+165.3% and +151.7%, respectively) in the three cell lines (Figure 6B).

## 4. Discussion

Although many studies emphasized the uncountable positive effects on human health of the phenolic compounds contained in certain edible or medicinal plants, only a few reports have investigated the possible influence of the gastrointestinal digestion on their efficacy. The main finding of our study using a simulated in vitro GI digestion is that the digestion boosters the antioxidant activity of *T. spicata* extracts, while it reduces the antiproliferative potential. We may attribute these differences in the biological activity of the extracts to the modifications in the phenolic profile caused by the simulated digestion.

To our knowledge, no studies have previously documented the possible changes in bioactivity of *T. spicata* extracts or preparations after transit in the gastrointestinal tract. In this context, we assessed the content of the main macromolecules and the phenolome in ethanolic and aqueous extracts from *T. spicata* leaves, and for the aqueous extract, we also distinguished between the low and high mw fractions.

Regarding the ethanolic extract, the simulated digestion led to a significant decrease in almost all components, while in the aqueous extract, it led to a reduction of only phenols and flavonoids. Although the simulated digestion reduced the carvacrol content in the ethanolic extract in absolute terms, carvacrol became the most abundant PC in relative terms. On the other hand, in the aqueous extract, the simulated digestion significantly increased the content of salvalonic acid, which became the most abundant PC in relative terms (Table 2). The increase in salvalonic acid content likely depends on the reduction in the RA upon digestion, taking into consideration the fact that salvalonic acid derives form condensation of two units of RAs, and this compound appears to be the precursor to many related salvianolic acid derivatives [32]. Therefore, we could hypothesize that the reduction in the rosmarinic acid (RA) upon digestion had been balanced by the increase in the SA.

We wish to emphasize as the biotransformation of parental phenolic compounds during the digestive process could mainly depend on the enzymes and the physiological environment of the GI tract (pH, temperature, and electrolytes) [33]. Indeed, Karas et al. [34] suggested that about 10% of the PCs remain undigested in the plant matrix, with only 90% of them being digested and released. However, the effects of digestion might vary according to the plant materials, and in the literature, we found two different outcomes: one stating the increase in phenolic compounds upon digestion [35], and the other one showing a reduction [36]. Indeed, our data are in accordance with reports showing a reduction such as those showing a reduced PC content in Brassica oleracea [37], as well as in Chilean white strawberry [38] upon digestion.

The idea is that the GI digestion may be unable to release all PCs, leaving a considerable amount of non-extractable polyphenols (NEPs) being trapped by dietary fibers, macromolecules (i.e., proteins), or polysaccharides through hydrophobic, hydrogen, and covalent bonds [39]. Therefore, NEPs reach the colon nearly intact [40]; however, only the phenolic components released from the matrix are absorbable from the GI barriers, and this could explain the enrichment in the secondary metabolites that we observed in the low mw fractions (mainly carbohydrates and proteins). NEPs may be released from the food matrix in the colon by the action of microbiota thus becoming bioavailable, absorbed, and bioactive [41], and this point specifically will inform our upcoming investigations. In conclusion, our findings indicate that the effect of digestion was greater on the ethanolic extract, and this was likely to due to degrees of solubility of different phyto-constituents.

In a previous paper [15], we demonstrated the great antioxidative potential of the T. spicata ethanolic extract, being higher than the aqueous one. Interestingly, the antioxidant potential of both the ethanolic and aqueous extracts was boosted by the simulated digestion. This could be related to the noticeable enrichment of each extract in terms of bioactive compounds, namely, carvacrol in the digested ethanolic extract, and salvalonic acid in the aqueous digested extract according to previous scientific reports [42].

Although polyphenols are generally considered as antioxidant compounds, at very high concentrations, they are known to play a prooxidant effect that might promote apoptosis, especially in highly proliferative cells such as cancer cells [43,44]. A previous study of our group reported remarkable antiproliferative and pro-apoptotic effects on tumor cell lines for the ethanolic extract from *T. spicata* when tested at a rather high concentration (100 mg/mL) [17]. In the present study, we verified that the in vitro cytotoxic activity of the ethanolic extract on cancer cell lines was maintained after the simulated digestion, but with lower efficacy compared with the crude extract. As widely reported, the antiproliferative activity of a plant is firmly correlated with the PCs [45]. Accordingly, we observed a decreased antiproliferative potency for the digested extract compared to the crude one, and this result parallels well with the reduced PC content upon digestion, in particular the reduction in carvacrol, which is the most potent antiproliferative agent in our study.

We evaluated the free radical production to clarify the mechanisms sustaining the anti-proliferative activity of the ethanolic extract before and after digestion. The results of the present study showed that the pro-oxidant property of the ethanolic extract was not only maintained after the digestion, but it was even bigger in terms of ROS production when compared to TE-dig. On the other hand, the NO release was higher in the crude extract compared to the digested one.

Interestingly, we observed that both the crude and digested ethanolic extracts were less potent in ROS production compared to carvacrol, while in terms of NO release, the extracts were more potent than carvacrol. This can indicate that the antiproliferative potential of *T. spicata* is exerted by acting as ROS-mediating apoptosis and inducing the release of cytotoxic mediators. Nevertheless, further studies should be performed at this level to have a clear idea about the mechanistic mode of action.

## 5. Conclusions

In summary, although we observed a reduction in the PCs and modulation in the phenolome of both ethanolic and aqueous *T. spicata* extracts upon the simulated GI digestion, the antioxidant activity was significantly enhanced. However, the antiproliferative potential of the ethanolic extract was reduced. Accordingly, we can come to an assumption that the digestion process had an impact on the nutritional value of *T. spicata*, but it kept its biological effectiveness.

As a final word, we can confidently say that *T. spicata* can represent a good and considerable source of PCs with potent antioxidant and antiproliferative bioactivities. In particular, the detected panel of bioactive compounds in *T. spicata* makes this edible plant a potent candidate to be used as a dietary supplement for different therapeutic purposes.

## Figures and Tables

**Figure 1 antioxidants-11-01778-f001:**
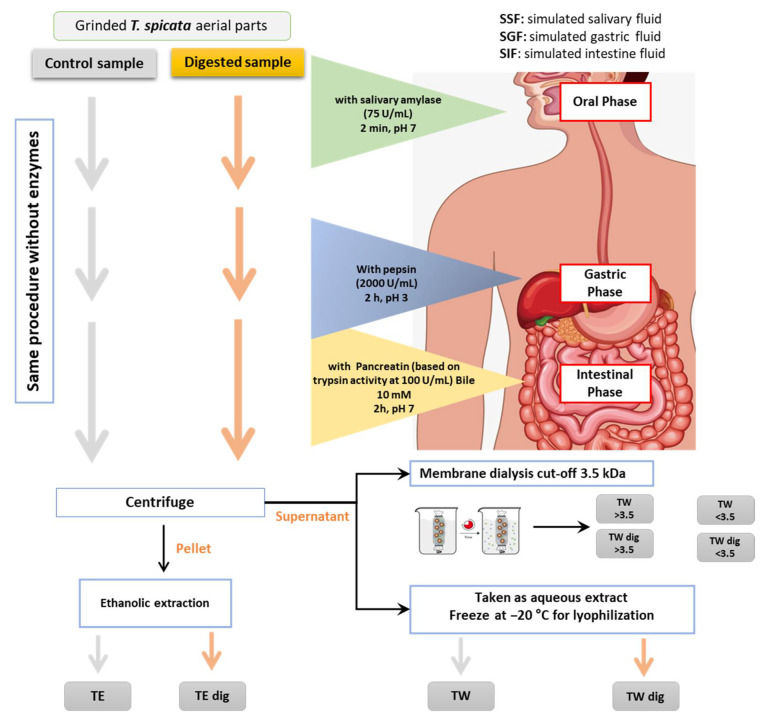
A schematic presentation describing the steps of the simulated digestion in the three phases: mouth, stomach, and intestine, in order to obtain the digested ethanolic (TE) and aqueous (TW) extracts. The same procedure was followed for preparation of crude extracts without the use of enzymatic digestion. The TW and TW dig were subjected to a membrane dialysis with a cut-off of 3.5 kDa to obtain low and high mw fractions.

**Figure 2 antioxidants-11-01778-f002:**
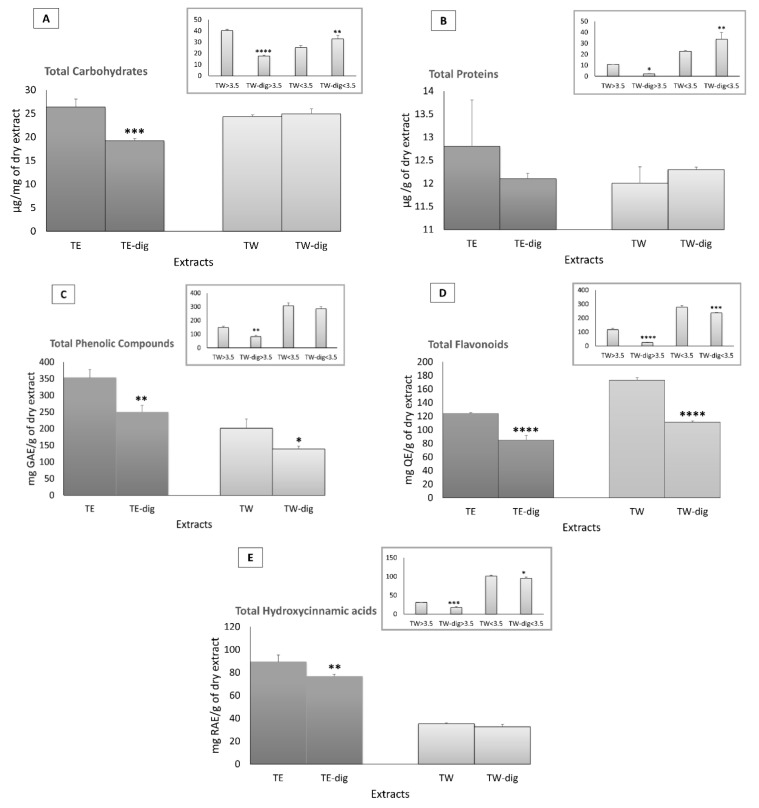
Quantification of total carbohydrate contents (TCC) (**A**), total protein content (TPrC) (**B**), total phenol content (TPC) (**C**), total flavonoid content (TFC) (**D**), and total hydroxycinnamic acid content (THAC) (**E**) of TW and TE, and dialyzed fractions before and after simulated in vitro digestion. All the contents were quantified spectrophotometrically and expressed as µg/mg of the dry extract, µg/g of dry extract, mg of gallic acid equivalent per g of dry powder extract (mg GAE/g dry extract), mg of quercetin equivalent per g of dry powder extract (mg QE/g dry extract), and mg of rosmarinic acid equivalents (RAE) per gram of dried weight extract (mg of RAE/g extract), respectively. Samples were measured in triplicate, and significant differences between digested and undigested extracts are denoted by symbols: * *p* < 0.05, ** *p* < 0.01, *** *p* < 0.001, and **** *p* < 0.0001.

**Figure 3 antioxidants-11-01778-f003:**
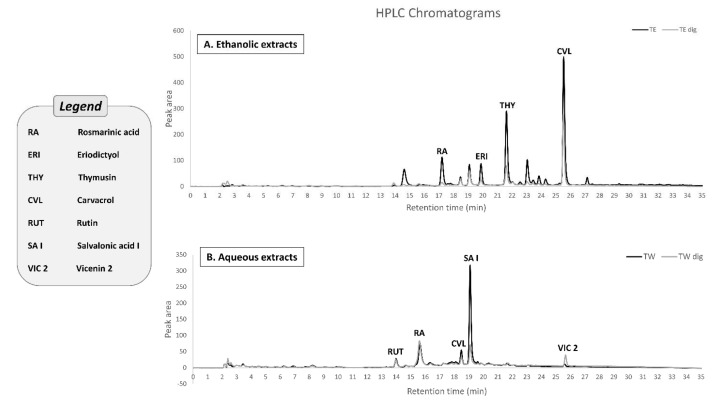
HPLC–UV chromatographic profiles for both ethanolic and aqueous extracts of Thymbra spicata before and after digestion wherein their pure polyphenols were recorded at 280 nm: (**A**) Chromatogram of the ethanolic extract (TE and TE-dig) showing the following peaks: 1: carvacrol; 2: thymusin; 3: rosmarinic acid; 4: eriodictyol. (**B**) Chromatogram of the aqueous extract (TW and TW-dig) showing the following peaks: 1: salvalonic acid I; 2: rosmarinic acid; 3: carvacrol; 4: vicenin; 5: rutin.

**Figure 4 antioxidants-11-01778-f004:**
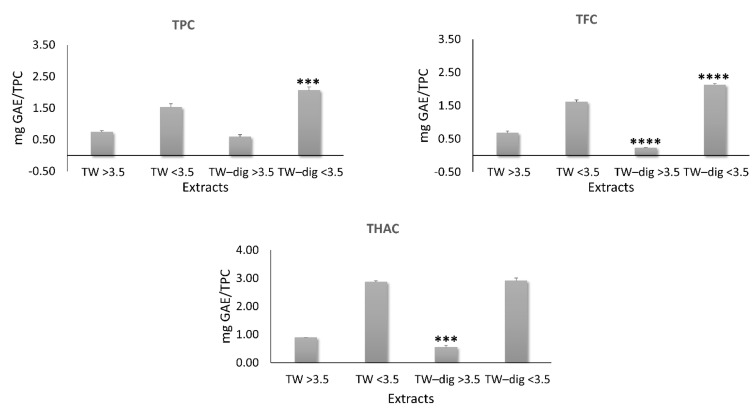
Normalized total phenol content (TPC), total flavonoid content (TFC), and total hydroxycinnamic acid content (THAC) of high and low mw fractions of the aqueous extract by the TPC of the corresponding extracts. Significant differences between digested and undigested extracts are denoted by symbols: *** *p* < 0.001 and **** *p* < 0.0001.

**Figure 5 antioxidants-11-01778-f005:**
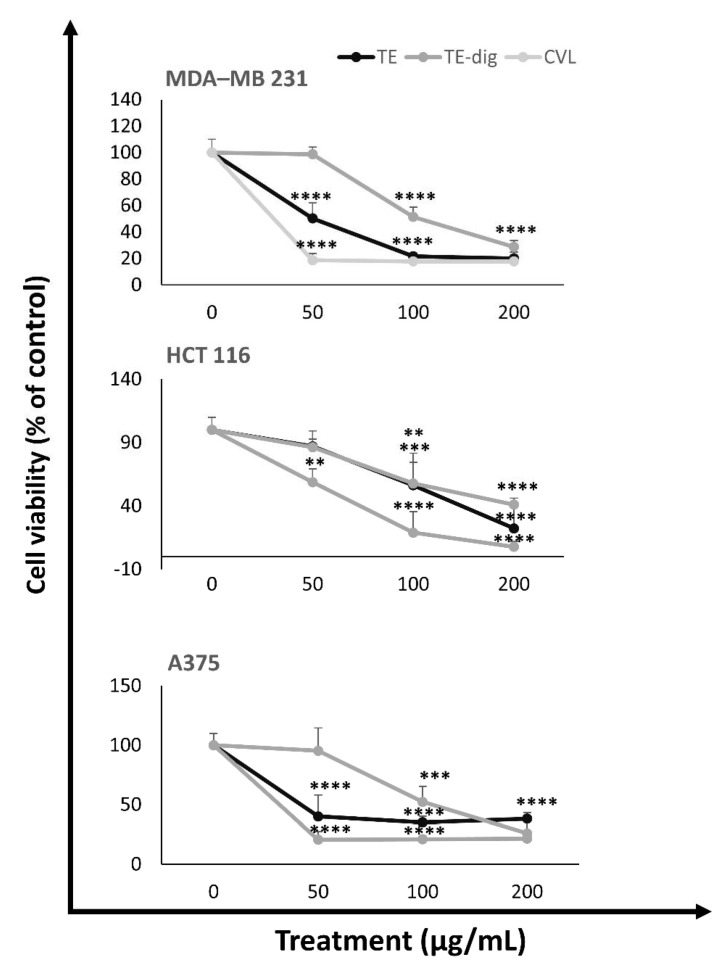
Antiproliferative activity of TE, TE-dig, and CVL on three representative human cancer cell lines: breast (MDA-MB 231), colon (HCT 116), and melanoma (A375) cells. The cell viability was expressed as a percentage (%) with respect to the control. Data represent the mean of at least five independent experiments. Statistical analysis for cell viability data was performed using two-way ANOVA followed by Tukey’s post-test ** *p* < 0.01, *** *p* < 0.001, **** *p* < 0.0001).

**Figure 6 antioxidants-11-01778-f006:**
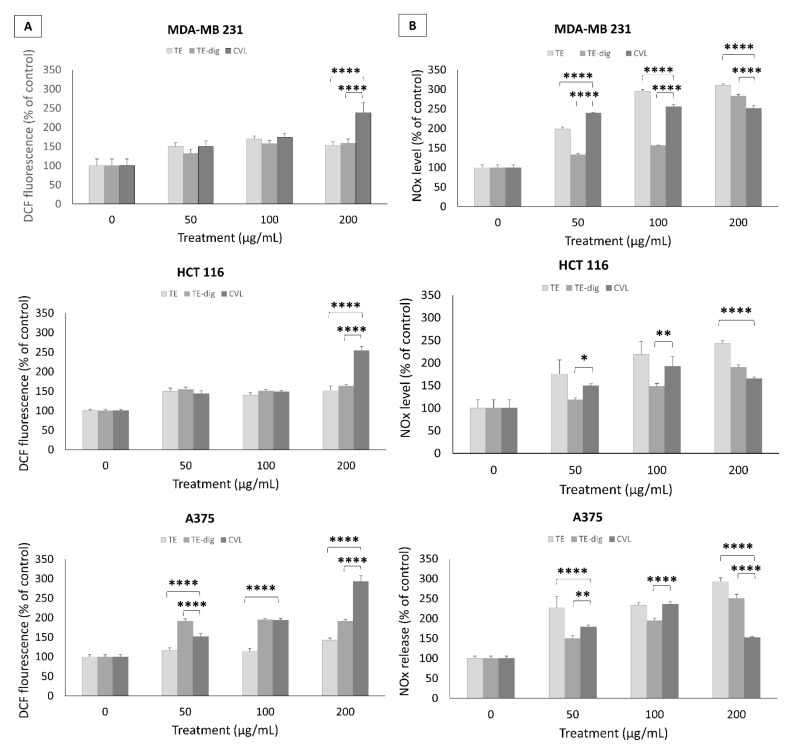
Pro-apoptotic effects of TE, TE-dig, and CVL on three cell lines: MDA-MB 231, HCT116, and A375, were assessed by measuring the ROS (**A**) and NO production (**B**) using spectrophotometric and fluorometric analyses, respectively. Values are expressed as % of control. Data represent the mean of five independent experiments. Statistical analysis for cell viability data was performed using two-way ANOVA followed by Tukey’s post-test (* *p* < 0.05, ** *p* < 0.01, **** *p* < 0.001).

**Table 1 antioxidants-11-01778-t001:** Chemical composition of the buffer employed in the simulated digestion. SSF: salivary fluid; SGF: gastric fluid; SIF: intestinal fluid.

Volume (mL)							
Simulated Digestion Fluid	pH	KCl (0.5 M)	KH_2_PO_4_ (0.5 M)	NaHCO_3_ (1 M)	NaCl (1.5 M)	MgCl_2_(H_2_O)6 (0.15 M)	Na_2_CO_3_ (0.5 M)
SSF	7	15.1	3.7	6.8	-	0.5	0.06
SGF	3	6.9	0.9	12.5	11.8	0.4	0.5
SIF	7	6.6	0.8	42.5	9.6	1.1	-

**Table 2 antioxidants-11-01778-t002:** Phenolic compounds identified in TE (A) and TW (B) before and after digestion using HPLC–MS/MS in the negative ionization mode.

**(A): Ethanolic Extract (TE)**								
**a**	**RT (min)**	**Measured *m*/*z***	**MS/MS Fragments**	**Proposed Compound**	**TE** **Area (%)**	**TE-Dig** **Area (%)**	**TE** **Peak Area**	**TE-Dig** **Peak Area**
1	14.1	593	575, 503, 473, 383, 353	Vicinin 2	0.34	2.89	46	137
2	14.5	303	285, 177, 125	Dihydroquercetin (taxifolin)	7.19	1.87	982	89
3	17.1	417	371, 287, 263	Eriodictyol derivative	8.69	2.84	1187	135
4	18.5	609	301	Rutin	2.04	4.04	278	192
5	19.1	359	223, 197, 179, 161, 133	Rosmarinic acid	5.73	10.53	782	500
6	19.7	287	269, 151, 135, 107	Eriodictyol	6.09	4.07	831	193
7	21.5	329	314	Thymusin	21.20	14.64	2894	695
8	23	285	257, 243, 151	Apiginin	0.85	0.63	116	30
9	22.8	269	201, 181, 149	Luteolin	6.64	2.40	906	114
10	23.3	343	328, 313, 300, 285	Unknown	0.97	0.93	132	44
11	24	165	149	P-Cymene-2,3-diol	2.23	1.14	305	54
12	24.3	343	328, 313	Cirsilineol	1.38	0.82	189	39
13	25.7	–	–	Carvacrol	34.81	52.94	4752	2513
14	27.3	329	314, 299, 286, 271	3,4,3′,4′-Tetrahydroxy-5,5′-diisopropyl-2,2′-dimethylbiphenyl	1.85	0.25	253	12
**(B): Aqueous Extract (TW)**								
**a**	**RT (min)**	**Measured *m*/*z***	**MS/MS fragments**	**Proposed Compound**	**TW** **Area (%)**	**TW-Dig** **Area (%)**	**TW** **Peak Area**	**TW-Dig** **Peak Area**
1	8.1	305	225	Gallocatechin	2.35	1.16	114	33
2	12.2	387	369, 225, 207, 163	Tuberonic acid glucoside	0.00	0.00	0	0
3	14	593	575, 503, 473, 383, 353	Vicenin 2	5.68	8.40	275	239
4	15	637	461, 351, 285	Luteolin-O-diglucuronide	0.00	0.00	0	0
5	15.4	537	493, 339	Salvalonic acid I	19.51	42.27	945	1203
6	15.7	477	397, 373, 343, 301	Quercetin-glucuronide	0.00	0.00	0	0
7	16.3	595	473, 429, 287	Eriodictyol-rutinoside	0.62	2.14	30	61
8	16.5	623	433, 287	Luteolin-glucuronide-hexoside	0.23	0.88	11	25
9	17	717	537, 519, 475, 365, 339	Salvalonic acid E\B	1.94	3.34	94	95
10	17.4	461	285	Luteolin 7-O-glucuronide	0.00	0.00	0	0
11	17.6	593	285	Luteolin-O-rutioside	0.00	0.00	0	0
12	17.9	441	418, 405, 373, 305, 225, 175	Unknown	0.00	0.00	0	0
13	18.1	521	359, 179, 161	Rosmarinic acid-glucoside	0.00	0.00	0	0
14	18.5	609	301	Rutin	7.80	7.48	378	213
15	19.1	359	223, 197, 179, 161, 133	Rosmarinic acid	57.41	18.80	2781	535
16	19.7	549	387	Tuberonic acid derivate	1.09	1.09	53	31
17	19.8	607	559, 427, 299, 284	Methyl kaempferol O-rutinoside	1.03	1.65	50	47
18	21.6	491	443, 311, 267	Salvalonic acid C	1.09	2.57	53	73
19	25.7	–	–	Carvacrol	1.24	10.22	60	291

**Table 3 antioxidants-11-01778-t003:** (A) The radical scavenging activity of the *T. spicata* extracts before and after digestion. Values are reported as Trolox equivalent (µg TE/mg dry extract). (B) Pearson correlation (two-tailed) between TFC, THAC, TPC, and antioxidant parameters (DPPH, ABTS, and FRAP). All values are mean ± SD from at least three independent experiments. Samples were measured in triplicate for each experiment. Significance is denoted by symbols: * *p* < 0.05, ** *p* < 0.01, and *** *p* < 0.001.

**(A) Radical Scavenging Assays**							
**Assays**	**DPPH**		**ABTS**		**FRAP**		**TPC**
**Samples**	**Trolox** **Equivalent ± SD**	**Trolox** **Equivalent/TPC ± SD/TPC**	**Trolox** **Equivalent ± SD**	**Trolox** **Equivalent/TPC ± SD/TPC**	**Trolox** **Equivalent ± SD**	**Trolox** **Equivalent/TPC ± SD/TPC**	
TW (total)	90 ± 12.3	0.447 ± 0.061	210.6 ± 32.4	1.046 ± 0.161	73.5 ± 15.6	0.365 ± 0.077	201.4
TW < 3.5 kDa	90.8 ± 9.8	0.295 ± 0.032	234 ± 25.9	0.761 ± 0.084	146.1 ± 18.9	0.475 ± 0.061	307.4
TW > 3.5 kDa	80.3 ± 7.9	0.540 ± 0.053	250 ± 32.4	1.681 ± 0.218	78.5 ± 10.8	0.528 ± 0.073	148.7
TW dig	83.8 ± 9.5	* 0.603 ± 0.068	173.7 ± 19.1	1.251 ± 0.138	62.8 ± 9.2	0.452 ± 0.066	138.9
TW Dig < 3.5 kDa	86.4 ± 11.1	0.302 ± 0.039	182.4 ± 17.5	0.637 ± 0.061	140.9 ± 16.1	0.492 ± 0.056	286.3
TW Dig > 3.5 kDa	65.1 ± 7.9	* 0.798 ± 0.097	250 ± 32.4	* 3.063 ± 0.397	42.3 ± 7.6	0.518 ± 0.093	81.6
TE	89.5 ± 12.1	0.254 ± 0.034	250 ± 32.4	0.708 ± 0.092	92.2 ± 6.1	0.261 ± 0.017	353
TE dig	94 ± 12.4	* 0.376 ± 0.05	210.7 ± 23.9	0.843 ± 0.096	88.3 ± 7.9	* 0.353 ± 0.032	250
(**B**) Pearson Correlation (Two-Tailed)							
		TFC	THAC	TPC	ABTS	DPPH	FRAP
TFC			0.6797	0.6274	0.3923	0.5646	** 0.8985
THAC		0.6797		*** 0.9375	0.4215	0.6726	** 0.8913
TPC		0.6274	*** 0.9375		0.5867	* 0.7612	* 0.7755

**Table 4 antioxidants-11-01778-t004:** IC50 values (50% cell viability inhibitory concentration) determined for *T. spicata* ethanolic extracts (TE and TE dig) and Carvacrol (CVL) in the three cell lines over a long-term (24 h) period under analysis.

Cell Lines	TE	TE-Dig	CVL
MDA-MB 231	58.447	112.103	23.278
HCT116	110.238	147.51	59.625
A375	31.443	107.067	19.912

## Data Availability

Datas are contained within the articles.

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
