# Peer review of "Influence of Simulated In Vitro Gastrointestinal Digestion on the Phenolic Profile, Antioxidant, and Biological Activity of Thymbra spicata L. Extracts"

_antioxidants, 2022, doi:10.3390/antiox11091778_

Round 1

Reviewer 1 Report

Review antioxidants- 1865334

Influence of simulated in vitro gastrointestinal digestion on the phenolic profile, antioxidant, and biological activity of Thymbra spicata L. extracts

The whole paper: I don't understand why chemical formulas are written without using subscript?

Line 20-21: rewrite the sentence. Obecnie szyk zdania I użyte wyrazy nie są najszczęśliwiej użyte

 One of possibility: Currently, it is still unclear whether and how the pattern of  phenolic compounds (PCs) found in the plants, as well as their bioactivity, could be modified during the gastrointestinal transit.

Line 23: occurring that occurs

Line 24: extracts from extracts of aerial parts of

Line 37: Rewrite keywords once again. Most of words now were already used in the title. Searching machines always search both title and keywords so it shouldn't be doubled. 

Introduction

The introduction contains a lot of interesting information from the literature on the chemical composition and properties of the plant. When discussing a species, please use the proper botanical notation. Nevertheless, a very important part of the experiment should be supplemented in the introduction, namely, there is nothing about the method.

Are there other simulation methods? What are the pros and cons of using simulation methods. Does the chosen method give a good picture of absorption by the body?

Materials and Methods

Lines 96-114. Please divide this fragment into subpoints. As it stands, it is difficult to keep up with a single procedure.

Lines 119, 209: units! Please check the whole paper

Please write the procedure in points that would be easier to follow. 

Line 112: For preparing to prepare 

Lines 121-122:  calibration curve of glucose  a glucose calibration curve

All procedures described in points 2.4-2.14 are written correctly, but due to their number and often significant difference I would suggest changing the way of writing. From where the term "briefly" is, present it with a diagram. It would make it very easy for the reader to follow and compare the procedures.

Results

Reading this chapter I have doubts about the purpose of the descriptions. Does the main purpose was to compare different extracts or to compare the characteristics of the extract before and after digestion. Too many values are given in the text with excessive accuracy. Also, the standard deviation and the p-value  always given darken the image very much. The tors should correct this and limit the number of digits and concentrate only on giving significant differences. If they necessarily want to give some numerical values to them, they can be given on the charts, the more that the possibility of reading these values exists and the presented significance of the differences speaks for itself without exact knowledge of the numbers. Giving a four-digit value in this case and a four-digit standard deviation is simply art for art's sake, not a representation of the result. A simpler description, the reader will be able to keep up with the line of thinking more easily and not have dubts  like me, what is the purpose of this chapter.

In materials and methods, the authors presented subsequent analyses in an orderly manner. I don't understand why the results description didn't follow the same order.

Disscusion

The discussion needs to be rewritten in its current form, the discussion does not meet the requirements of this type of part. The works that are compared are far too  few and therefore there is no discussion with the achievements of other scientists and this is important in the course of something like another description of results and achievements.

Please add the conclusions.

Reviewer 2 Report

Submission Antioxidants-1865334 (“Influence of simulate in vitro gastrointestinal digestion on the phenolic profile, antioxidant, and biological activity of Thym-3 bra spicata L. extracts”) intends to describe the chemical composition, stability, and biological activity of phenolic compounds from plants, after in vitro digestion. The manuscript present relevant information for Antioxidants’ readers, although not much novelty in the scientific approach is presented.

In my opinion there are some relevant information’s that are missing while other is not presented in a clear way.  This clarification should greatly improve this manuscript and the conclusions that were obtained. For instance, the authors attributed the differences in the biological activity of the extracts to the modifications in the phenolic profile caused by the simulated digestion. However, this profile modification is only explored superficially.

Additionally, throughout the text are several misspelling and formatting errors that should be revised.

In my opinion major revisions are needed before manuscript acceptance. General comments are listed below:

Keywords: Please simplify keywords: Use simple and meaningful words;

Introduction:

Line # 40: Please rewrite first sentence to: “Dietary phytochemicals are found abundantly in fruits, vegetables, grains, plant-based foods and beverages”;

Line # 60: Please replace “initial food” for “parental phenolic compounds”;

Line # 61: in vitro; please correct within the manuscript.

Line # 80: The last sentence is related to results and conclusions and should not be presented in the introduction section.

Materials and Methods:

Line # 85: Please detail reagents origin. This is particularly relevant for enzymes, in which the authors should dive information regarding enzymes activity, origin, etc;

Table 1: Numbers in the chemical formula should be in subscript; Also, in the following lines, the authors need to correct this misspelling;

Line # 98: Missing a space in “stock75”;

Line # 100: Missing a space in after the volumes added;

Line # 101: For the gastric digestion phase, the authors used 90 mL of the oral phase. However, from the information reported for the oral phase, the authors should only have 34 mL from the oral phase. Did the authors intended to write 9.0 mL, instead? Please clarify this information.

Line # 105: How did the authors stopped the digestion? Were enzymes inactivated? Please clarify how the authors performed;

Line # 107: What were the ethanolic extract conditions? Were the extracted samples lyophilized?

Line # 110: Please add information regarding dialysis. In which volume were the samples dialyzed, for how long, how where the low mw samples treated? Please add this information;

Line # 111: Missing a space in molecular weigh;

Line # 112: Meaning of mw and again, missing a space in > 3.5;

Line # 113: For preparing the undigested aqueous and ethanolic extracts, the same procedure was followed for extraction without the first part of enzymatic digestion. Did the authors subjected the samples to thermal treatment (37º C) and to the different pH solutions as for the digested samples, only excluding enzymes? Or was it only performed a phenolics extraction? Was this extraction only performed to characterize the initial phenolic composition? Please, clarify this information;

Line # 133: Information regarding UV-vis microplate reader should be added to the previous sections;

Line # 224: Replace l for l;

Results:

Line # 244: explanation why to assess the low and high molecular weight fractions should be presented in the manuscript. Why was this information relevant for this work?

Line # 247: statistics should be > or < (p value);

Figure 2: The insets in figure 2 are not visible (axes titles). Please increase this figure for proper reading;

Line 259: In the water extract the phenolic compounds amount is higher in the dialyzed samples compared to the crude extract. Is this data ok?

Line # 277: There is some information repeated regarding phenolics composition. Please correct this information;

Line # 289: How did occur and enrichment in carvacrol after digestion? Did it resulted from other phenolic compounds that may have been metabolized during digestion? Or could they be liberated from plants solid parts, thus increasing their amount in the extract?  Please clarify this information;

In Figure 3 is not perceptible the chromatogram related to the digested samples. Please, also explain the abbreviation within the figure and add the numbers presented in the figure caption;

Line # 326: In DPPH assay, water extract presents a significant higher antioxidant capacity, although the lower amount in phenolic compounds; Should this be expected? Please explore this result;

Table 2: Please add commas between obtained MS7/MS fragments;

Figure 4: Phenolics normalization was performed with phenolic compounds amount in the crude water extract? Please clarify this information;

Table 3: Please rewrite table caption. Caption should be shelf-explained. Remove the second sentence. Units are not the same as presented in the materials and methods section. Please clarify this. Remove the dot from SD; p should be < and not ≤; Please add the “a” meaning in the table;

Regarding the information presented in this table, TPC is not similar to do ones presented in the text. Is this data corrected?

Figure 5: Please use the same format in the graphic axis;

Line # 374: Please remove the information in parenthesis. This information has already been presented throughout the text;

Table 4 caption is missing a dot; Some of the data within this table is in bold. Is this a mistake or please explain why this data was highlighted;

Line # 382: Add RA meaning;

Line # 388: the authors suggest that phenolics biotransformations may be due to the physiological environment. Did the authors performed a digestion without enzymes, considering only pH value,  temperature, .. and assessed phenolics biotransformation? This should greatly improve the discussion of the results.

Line # 392 and 395: remove the extra space;

Line # 399: non-extractable polyphenols should be essentially linked to polysaccharides besides proteins;

Round 2

Reviewer 1 Report

After making changes, the work in my opinion. It is definitely better. The authors have taken my comments into account satisfactorily. In addition to the changes in the whole document, my script added part of the conditions. However, I would suggest minor changes in the first and third sentences.  In my opinion, they should read as follows. The work is suitable for publication in the journal Antioxidants.

part of conclusions with suggested changes:

In summary, although we observed a reduction in PCs and modulation of phenolomes of both ethanolic and aqueous T. spicata extracts upon simulated GI digestion, the antioxidant activity was significantly increased. .... Consequently, we can assume that the digestion process had an impact on the nutritional value of T. spicata, but it kept its biological effectiveness. ...